# Effect of La_2_O_3_ on Microstructure and Thermal Conductivity of La_2_O_3_-Doped YSZ Coatings

**DOI:** 10.3390/ma12182966

**Published:** 2019-09-12

**Authors:** Xiaojie Guo, Chucheng Lin, Jimei Zhang, Ziwei Liu, Caifen Jiang, Wei Zheng, Yi Zeng

**Affiliations:** 1The State Key Lab of High-performance Ceramics and Superfine Microstructure, Shanghai Institute of Ceramics, Chinese Academy of Sciences, Shanghai 200050, China; guoxiaojie@mail.sic.ac.cn (X.G.); chucheng@mail.sic.ac.cn (C.L.); jmzhang@mail.sic.ac.cn (J.Z.); ziweiliu@mail.sic.ac.cn (Z.L.); cfjiang@mail.sic.ac.cn (C.J.); zhengwei@mail.sic.ac.cn (W.Z.); 2Center of Materials Science and Optoelectronics Engineering, University of Chinese Academy of Sciences, Beijing 100049, China

**Keywords:** thermal conductivity, recrystallized grains, melting, grain size

## Abstract

Enhancing the properties of thermal barrier coatings (TBCs) by doping with rare earth elements has been a hot topic for a while. La_2_O_3_ and Y_2_O_3_ co-doped ZrO_2_ (La-YSZ) TBCs and yttria-stabilized zirconia (YSZ) TBCs were deposited by atmospheric plasma spraying (APS), and the comprehensive effects of La^3+^ on the microstructure and property were investigated. The thermal conductivity and microstructure were investigated and were compared with YSZ. The recrystallized fraction components of all TBCs were quantified. It is clearly found that the component of “recrystallized” and “deformed” grains for La-YSZ TBCs is much higher than that for YSZ TBCs. This could be due to La^3+^ doping enlarging the lattice parameter of YSZ and thus increasing the melting index, which in turns leads to the smaller grain size of La-YSZ TBCs. As a result, the thermal conductivities of La-YSZ TBCs were distinctly lower than those of YSZ TBCs.

## 1. Introduction

Yttrium oxide-stabilized zirconia (YSZ), a high-temperature structural material with low thermal conductivity and high thermal expansion, is widely used as a thermal barrier coating (TBC) [1,2,3]. TBCs, having lower thermal conductivity, are widely used in turbine blades, which are usually operated at very high temperatures [4,5,6,7]. Engines operating at high temperatures require materials with lower thermal conductivity in order to insulate the other parts of the engine from hot burning gases [8,9,10,11,12]. In addition to the Y_2_O_3_-ZrO_2_ (YSZ) TBCs, it has been found that lanthanum-doping can effectively reduce the thermal conductivity of YSZ TBCs. It has been reported that La_2_O_3_ and Y_2_O_3_ co-doped ZrO_2_ (La-YSZ) coatings have lower thermal conductivity than YSZ, and reducing thermal conductivity by multiple doping has been a hot topic. Thus, there are many studies on La-YSZ TBCs [13,14,15,16].

Liu et al. investigated the microstructures of La_2_O_3_-modified YSZ TBCs that were deposited by atmospheric plasma spraying. It was reported that the grain growth of coatings under high temperatures can be improved by the addition of La_2_O_3_. They argued that lanthana particles segregated along the zirconia grain boundaries and that the trivalent and large cation strongly suppressed the mobility of grain boundaries and, consequently, grain growth [13]. However, there is no solid solution formed due to the separation of La_2_O_3_ and YSZ. In a more in-depth study, La_2_O_3_ co-doped Y_2_O_3_ stabilized ZrO_2_ powders were prepared by sol-gel method, chemical co-precipitation and calcination, then the thermal conductivity was calculated. Some Zr^4+^ ions were replaced in the matrix by La^3+^, causing a significant decrease in thermal conductivity of the TBCs. The reason for the decrease of thermal conductivity was proposed to be the reduction of the phonon mean free path caused by defect scattering and oxygen vacancies [14,15]. However, the influence on the microstructure of doping was not discussed in detail. Rauf et al. agreed that the substitution of Zr^4+^ with La^3+^ creates additional oxygen vacancies to maintain the neutrality of the lattice, and thus enhances phonon scattering. Moreover, they described the microstructure characterized by scanning electron microcopy (SEM) and transmission electron microscopy (TEM), and found that the doping of La^3+^ caused thinner splats and smaller grains. This increases the number of interfaces, thereby enhancing phonon scattering and, as a result, reduces thermal conductivity [16]. 

It is widely accepted that it should be the microstructure of coatings that strongly affect thermal conductivity. As a consequence, there are specific impacts of La-doping on microstructures such as pores, cracks and the grain size. Much of the literature had focused on pores and cracks, which were mainly caused by the spraying parameters. However, very few reports mentioned the influence of La^3+^ on crystal structures, such as grain size. Furthermore, for YSZ coatings, the effect of microstructure on the thermal conductivity has been studied extensively using the electron back-scattered diffraction (EBSD) method, but there are almost no reports regarding the effect of microstructures on the thermal conductivity of La-YSZ by EBSD method [17,18,19].

In this study, TBCs were plasma sprayed using La-YSZ and YSZ powders as feedstock. The grain size distribution is characteristic by SEM and EBSD. The recrystallized fraction component was quantified to better understand the effect of La-doping on grain size and thus the effect on thermal conductivity.

## 2. Materials and Methods

### 2.1. Materials and Preparation

The ZrO_2_-7 wt.% Y_2_O_3_ (commercial grade) and ZrO_2_-7 wt.% Y_2_O_3_-2 wt.% La_2_O_3_ (La-YSZ, Shenyang Shihua Weifen Materials Co. Ltd., Shenyang, China) powders were deposited on aluminum substrate (130 mm × 85 mm × 3 mm) using a Metco A-2000 atmospheric plasma spray equipment having F4-MB plasma gun (Sulzer Metco AG, Wohlen, Switzerland). The particle size and morphology of the raw materials are shown in Figure 1. It can be clearly found that the medium size (D50), D10 and D90 of the La-YSZ and YSZ powders are approximate. La-YSZ powder was used to fabricate three coatings, denoted as N1, N2 and N3, while YSZ was used to fabricate N4. Table 1 shows the different spray parameters for depositing the free-standing coatings. All the spray parameters were optimized in our previous research.

### 2.2. Microstructures of Coatings

All the TBCs were removed from the substrate and the free-standing coatings were mechanically shaped into pieces (3 mm × 3 mm × 1 mm) and then polished by triple ion beam polishing instruments (Leica EM TIC 3X, Wetzlar, Germany). Phase composition of the specimens was identified by X-ray diffraction technique (XRD, D8-Advance, Bruker, Hamburg, Germany; Cu Kα radiation, λ = 0.15406 nm). Cross-sections of the sample were investigated by SEM (Magellan 400, FEI, Hillsboro, OR, USA). Moreover, the grain size distribution was evaluated by EBSD (INCA SERIES, Oxford Instrument, Oxford, UK). The grain size and recrystallized fraction component were quantified using CHANNEL 5 software. Notably, noise reduction should be performed carefully with EBSD and the zero solutions were not taken into account. In the present work, only isolated points that were incorrectly indexed (i.e., wild spikes) were removed and then filled in using their seven neighboring points.

### 2.3. Thermal Conductivity Measurements

The thermal diffusivities were measured using commercial laser flash equipment (TD-79A, SIC, Shanghai, China) from room temperature to 800 °C (heating rate = 10°/min). The specific heat capacity (C_p_) of the coatings was measured by differential scanning calorimetry (DSC) (SIC; Perkin Elmer, Waltham, MA, USA) from room temperature to 800 °C and densities (ρ) were measured using Archimedes methods. The thermal conductivity was calculated by:$$\lambda =\alpha \rho C_{p}$$
where α, ρ and C_p_ are the thermal diffusivity, density and specific heat of the coatings, respectively.

## 3. Results and Discussion

### 3.1. Morphology and Phase Analysis

X-ray diffraction patterns of the coatings (N1, N2, N3 and N4) are shown in Figure 2. Mainly a tetragonal phase can be clearly found, and no other peaks appeared; showing that Zr^4+^ were partly replaced by La^3+^ and Y^3+^, and a substitutional solid solution formed for all the coatings [20].

Figure 3 shows micrographs of cross-sections of the N1 and N4 coatings. The thickness of N1 and N4 are 1.35 mm and 1.62 mm, respectively. The columnar grains were solidified from the melted fraction of La-YSZ (or YSZ) powder and the spherical-similar microstructure was retained from the unmelted La-YSZ (or YSZ) powder. For both coatings, almost no large and spherical pores exist, and the coatings have different sized grains. The thickness of the splat and the width of the column grains of N1 are distinctly smaller than N4, which indicates that La-YSZ powder was at a better molten state when deposited on the substrate than YSZ powder.

### 3.2. Thermal Conductivity of Coatings

To study the thermal conductivity of La-YSZ coatings and YSZ coatings, the thermal diffusion coefficient, density and specific heat were measured. The calculated thermal conductivities are shown in Figure 4. The thermal conductivity of the compound first decreased with the increase of the temperature from room temperature to 700 °C and in this process, small fluctuations appeared due to the stability of the instrument. Then, above 700 °C, the thermal conductivities increased, because the thickness of the coatings was insufficient, and thus the laser penetrated the coatings directly. The results indicate that the thermal conductivity of N1, N2 and N3 are all lower than that of N4 when the temperature is higher than 500 °C. At 700 °C, for which the accuracy of the test results can be guaranteed, the thermal conductivities of N1, N2, N3 and N4 are 0.87 W/mK,1.04 W/mK, 0.97 W/mK and 1.12 W/mK, respectively. In addition, the thermal conductivity of N1 is 22% lower than N4. This may be due to the fact that the doping of La reduces the grain size of the coating, and that the influence of La-doping in grain size is different for the three spraying parameters. The main reasons will be discussed in detail below.

The polished cross-section of TBCs was analyzed by EBSD, and the bond contract of all the coatings is shown in Figure 5. More than 85% of grains are exactly indexed, except for the pores and cracks. Five EBSD images were taken for each sample. It is worth noting that zero solution of YSZ coating, which might correspond to the porosity of coating, was higher than those of other coatings. In contrast, the thermal conductivity of YSZ coating was much higher than the others. This distinctly indicates that the effects of pores on the thermal conductivity might not be predominant.

The distributions of grain size for the four coatings are shown in Figure 6a. Considering that the thermal conductivity of N1, N2 and N3 is lower than that of N4, the effect of grain size on thermal conductivity of these four coatings is noticeable. The histogram in Figure 6b contains the aspect ratio of grains with the aspect ratio larger than 2 (AS > 2) and smaller than 2 (AS < 2) within the range of d < 0.9 μm. The AS is defined according to the definition: the longest axis divided by the shortest axis of the grain [21]. The area ratio of grains with d < 0.9 μm of N1, N2 and N3 are 87%, 86% and 83%, respectively. These are all higher than N4 (64%), and especially in the range of d < 0.9 μm, the area ratio of grains with AS > 2 of N1 is 53.2%, which is far higher than N4 (19.1%). Thus, it is easy to reach the conclusion that the melting state of N1, N2 and N3 is better than that of N4. Considering that the particle sizes of YSZ and La-YSZ are similar, the reason may be attributed to that the ionic radius of La^3+^ is 1.06 Å, while the ionic radius of Zr^4+^ and Y^3+^ are 0.89 Å and 0.72 Å, respectively; indicating that the lattice parameter of La-YSZ should be larger than that of YSZ. Moreover, there is a positive relationship between the melting index and lattice parameter [22]. Therefore, it is proposed that the in-flight melting state of La-YSZ is better than that of YSZ, and thus the grains with thinner splat and smaller size than YSZ can be obtained. The melting state of grains was characterized by a statistical recrystallized fraction component.

The recrystallized component of grains, which has been widely used in the research of metal materials [23], was first applied to quantify the molten status of spraying particles. It was mainly due to the fact that the plasma-spraying process is more similar to the quenching process. The particles are generally melted in the plasma torch, then deposited on the substrate and rapidly solidified. As a result, many defects are created having variable amounts in different grains. According to the internal average misorientation angle within the grain, “recrystallized”, “substructured” and “deformed” grains can be clearly defined and discriminated.

The recrystallized fraction component is used to detect the deformed and recrystallized grains, as shown in Figure 7. Firstly, “recrystallized” grains can be grouped into molten particles, because they have misorientations that are smaller than the defined angle. In addition, the “deformed” grains can also correspond to the molten particle that has undergone a fast melting and cold processing during plasma spraying, which will inevitably lead to lots of defects and deformation. In contrast, “substructured” grains would be considered to be unmolten or part-molten particles. The main reason could be that the unmolten particle causes a sub-grain boundary, although the whole internal average misorientation was smaller than that of the “deformed” grains. 

In these maps, the “recrystallized”, “deformed”, and “substructed” grains are indicated in blue, red, and yellow, respectively. The higher component of recrystallized and deformed grains in TBCs indicates the better molten state of the particles during spraying. It is obvious from Figure 8 that the melting states of N1, N2, and N3 are all better than for N4 because the component of the substructured grains for N4 is 63.4%, which is much higher than that for N1 (32.9%), N2 (35.6%) and N3 (27.5%). This indicates that La^3+^ doping of YSZ powders facilitates an improved melting of in-flight particles, which reduces the grain size of TBCs and, ultimately, the thermal conductivity is reduced.

## 4. Conclusions

La-YSZ TBCs and YSZ TBCs were deposited by the APS method; thermal conductivities were measured and the microstructures were characterized by SEM and EBSD. The thermal conductivities of La-YSZ TBCs were lower than those of YSZ TBCs. The relationship between La-doping and melting index and the effect on grain size were discussed in detail.

The main reason for lower thermal conductivity is attributed to the smaller grain size of La-YSZ TBCs, especially the narrower width of the grains.The relative amount of “recrystallized” and “deformed” grains for La-YSZ TBCs is much higher than that for YSZ TBCs. Thus, the reduction of grain size for La-YSZ TBCs is due to the increase of the melting index of grains, which attributed to the enlargement of the lattice parameter caused by La^3+^ doping.

## Figures and Tables

**Figure 1 materials-12-02966-f001:**
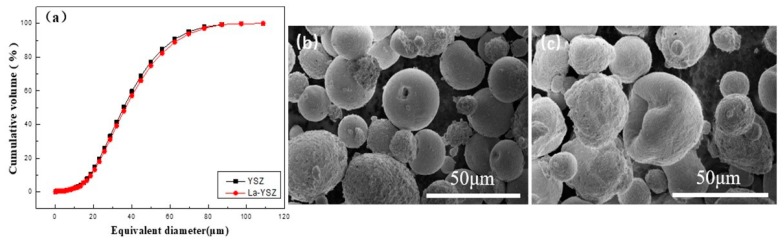
(**a**) Equivalent diameter distribution of powders, morphologies of (**b**) Yttrium oxide-stabilized zirconia (YSZ) powders and (**c**) La-YSZ powders.

**Figure 2 materials-12-02966-f002:**
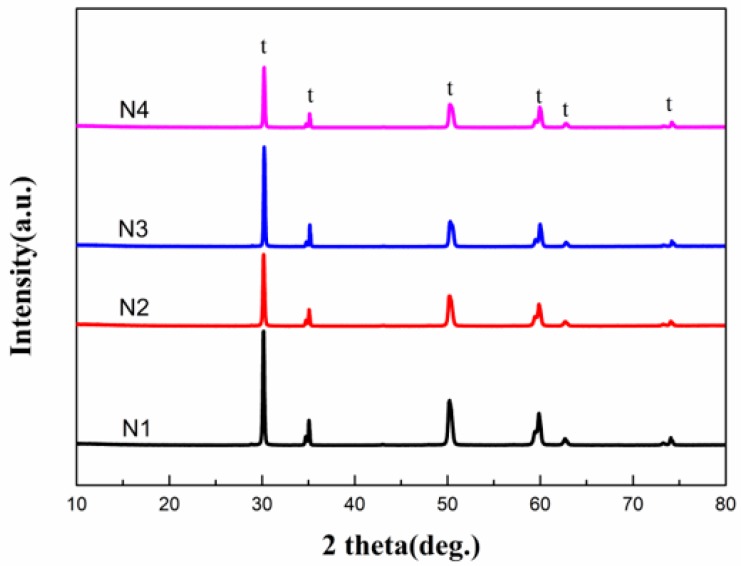
XRD patterns of samples.

**Figure 3 materials-12-02966-f003:**
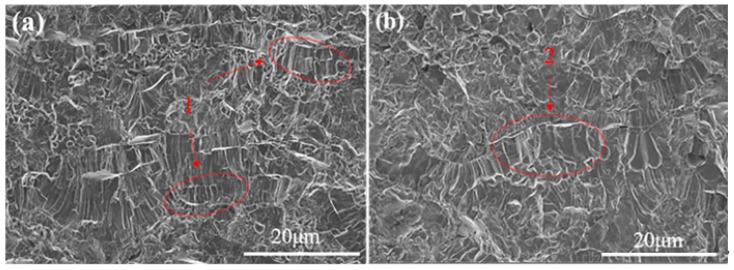
SEM micrographs of cross-sections of the coatings (**a**) N1 and (**b**) N4.

**Figure 4 materials-12-02966-f004:**
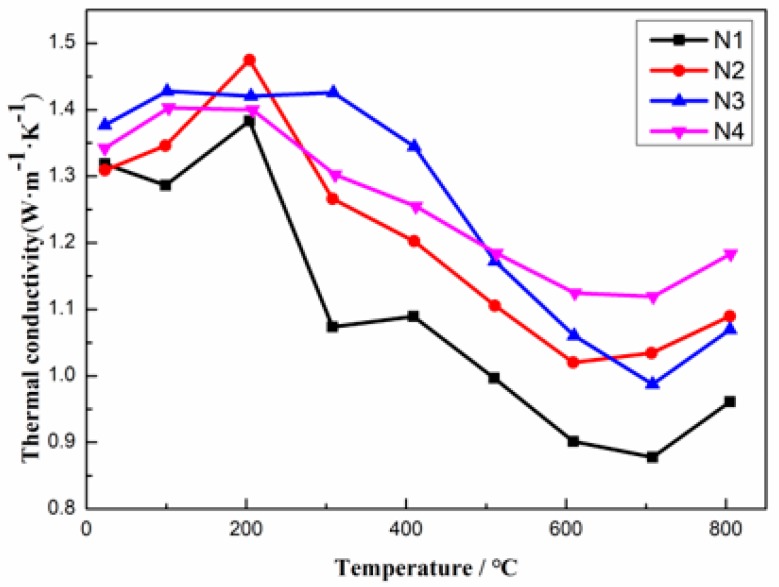
Thermal conductivities of N1, N2, N3 and N4.

**Figure 5 materials-12-02966-f005:**
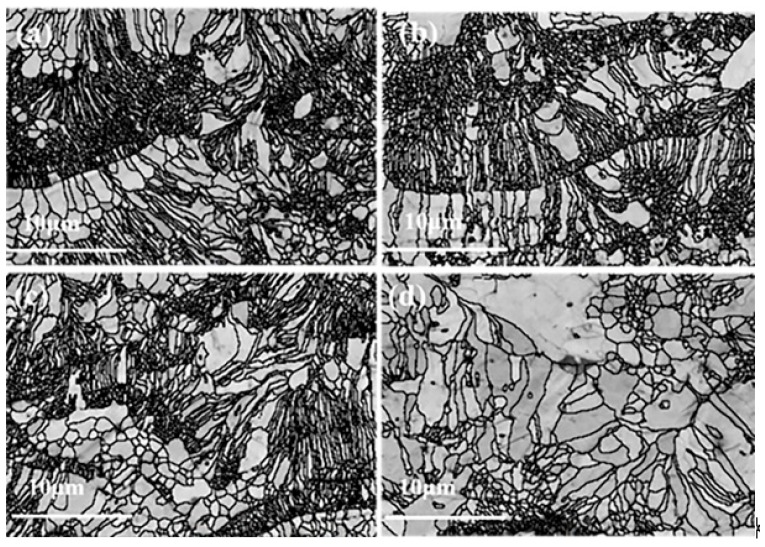
Bond contract of (**a**) N1, (**b**) N2, (**c**) N3, and (**d**) N4.

**Figure 6 materials-12-02966-f006:**
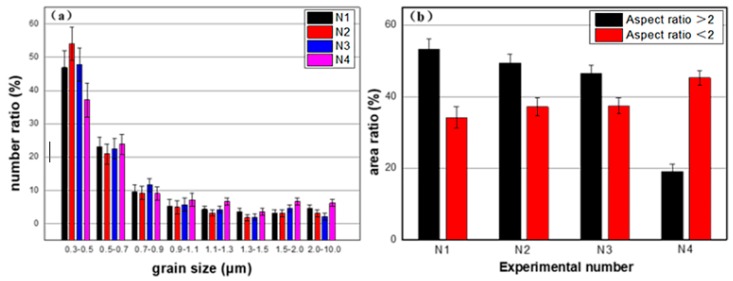
(**a**) Distribution of grain size of four coatings and (**b**) Area ratio of grains with different aspect ratio.

**Figure 7 materials-12-02966-f007:**
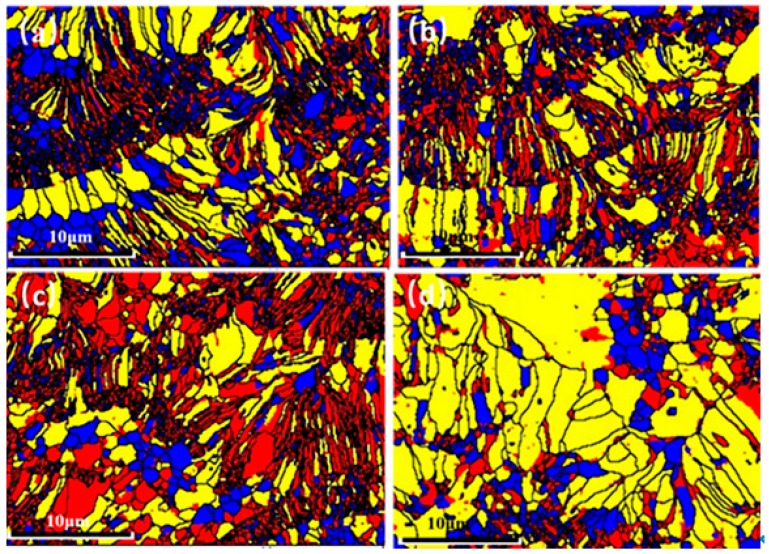
Recrystallized fraction component of (**a**) N1, (**b**) N2, (**c**) N3 and (**d**) N4.

**Figure 8 materials-12-02966-f008:**
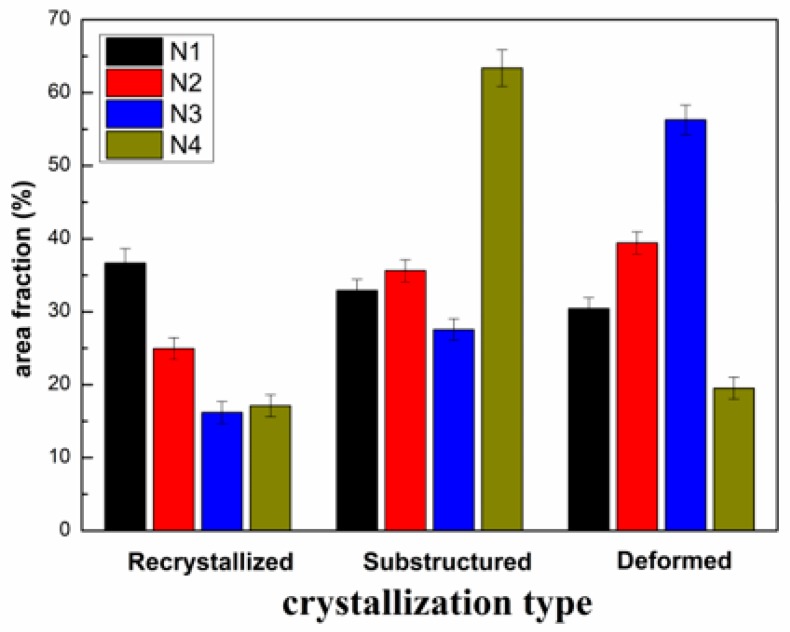
Area fraction of “recrystallized”, “substructured”, and “deformed” grains of four coatings.

**Table 1 materials-12-02966-t001:** Plasma spray parameters.

Spraying Conditions	N1	N2	N3	N4
Current (A)	600	550	650	600
Ar (L/min)	36	36	36	36
H_2_ (L/min)	12	12	12	12
Velocity (m/s)	200	208	213	211

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
