# Peer review of "Effect of La2O3 on Microstructure and Thermal Conductivity of La2O3-Doped YSZ Coatings"

_materials, 2019, doi:10.3390/ma12182966_

Round 1
Reviewer 1 Report
Dear Authors,
I suggest to Editors to accept this article, but minor revision, is necessary. In bullet points:
[Row 24] In "1. Introduction", could be useful to readers more details about La and TBCs and their applications (in general terms); [Row 67] In "2. Materials and methods", is it possible to see D10, D50 and D90 LS graph (particle distribution)?; [Row 73] Always in "2. Materials and methods", which is the angle between the injector and the flow, in the APS gun?; [Row 103] In "3. Results and discussion", is it possible to know the thickness of both coatings (less or more)?; [Row 197] In "References", could be useful more bibliography.Author Response
Dear reviewer:
Thanks for your kind suggestion. The corrections have been performed accordingly and shown in PDF File, please see the attachment.
2019/09/07

Reviewer 2 Report
Please see details of questions and remarks in the attached file (sticky notes).

Author Response
Dear reviewer:
Thanks for your kind suggestion. The corrections have been performed accordingly and shown in PDF file. Please see the attachment.
2019/09/07
